# Are Late-Born Young Soccer Players Less Mature Than Their Early-Born Peers, Although No Differences in Physical and Technical Performance Are Evident?

**DOI:** 10.3390/sports11090179

**Published:** 2023-09-08

**Authors:** Eduard Bezuglov, Georgiy Malyakin, Anton Emanov, Grigory Malyshev, Maria Shoshorina, Evgeny Savin, Artemii Lazarev, Ryland Morgans

**Affiliations:** 1Department of Sports Medicine and Medical Rehabilitation, Sechenov First Moscow State Medical University, 119991 Moscow, Russia; e.n.bezuglov@gmail.com (E.B.); kaisough@yandex.ru (M.S.);; 2High Performance Sports Laboratory, Sechenov First Moscow State Medical University, 119991 Moscow, Russia; 3Department of Internal Medicine, Mount Sinai Hospital, Chicago, IL 60608, USA; lazarevartemii1@gmail.com

**Keywords:** relative age effect, biological maturity, talent identification, soccer

## Abstract

The aim of the study was to compare the status of somatic maturity, anthropometry, strength, speed, and soccer-specific technical skills of players from leading youth soccer academies born in different quarters of the same calendar year. A total of 678 young male soccer players from eight leading Russian soccer academies took part in the study. The following anthropometric measures and physical characteristics were measured: height, weight, body mass index, countermovement jumps (CMJ), 5, 10, and 20 m sprints, speed dribbling, foot and body ball juggling, and short and long pass accuracy. The determination of somatic maturity as a percentage of projected adult height was collected. All subject dates of birth were divided into four quartiles according to the month of birth. The analysis of all data obtained was conducted both within the total sample and by quartiles of birth, according to the age group categories of 12–13 years, 14–15 years, and 16–17 years and the degree of somatic maturity. There was a widespread relative age effect, with 43.5% of early-born players and only 9.6% of late-born players representing the sample. Early-born players were more mature than late-born players (*p* < 0.001 and *p* < 0.001) but had no statistically significant differences in strength, speed, or soccer-specific skills.

## 1. Introduction

In recent decades, an important task for elite soccer academies has been to optimise the selection process. This may be due to both the high financial cost involved in training young players and the need to develop as many elite players as possible by the age of 18 capable of playing for the first team or selling to another club [1]. Therefore, it would be logical for elite soccer clubs to introduce talent identification programmes, which are based on a thorough assessment of players’ qualities in training and matches over a relatively prolonged period of time. It is also necessary to create a positive perception of the development environment for players [2,3]. The example of the German nationwide talent development programme clearly demonstrates the effectiveness of a multi-dimensional approach that integrates objective and subjective assessments during the selection process of talented young soccer players [4].

However, so far, the primary selection in elite youth soccer academies is often based on the subjective opinion of coaches as well as data from various physical and cognitive skills tests, the results of which are often assessed without regard to biological maturity status. In this context, a widespread relative age effect in highly competitive youth soccer is not surprising due to the traditional grouping of young players by a strictly defined time period (for example, 1 January to 31 December) [5,6,7,8,9,10]. Thus, children born at the beginning of this period (i.e., January or February) will be chronologically older than children born in October to December and therefore potentially more physically mature. Furthermore, these early-born children may also have a greater experience of different socio-cultural factors and, arguably, a physical and technical advantage. This leads to an over-representation of children born in the first half of the year in elite-level soccer academy teams [11,12,13].

Chronologically older soccer players have the potential to gain an advantage, not only during the primary selection process but also during the growth spurt period (12–15 years). This is likely to start earlier for the chronologically older soccer players and, consequently, may provide an advantage due to higher testosterone levels, which start to increase dramatically around 12 years of age in boys. Testosterone is a contributing factor in providing a physical advantage in strength, speed, endurance, and a range of psychological factors important for sporting success [14]. In other words, selection into the most highly competitive youth soccer academies is based on a maturation gradient [15].

Furley et al. provided confirmation that coaches may be subconsciously guided during their selection assessment by the body size of young athletes and reported a significant automatic link between body size and coaches’ assessment of talent, i.e., once again demonstrating that the concept of “sporting talent” is partly grounded in the perception of physical height amongst youth sport coaches [16]. Moreover, according to several studies, soccer coaches favour more early-born players than late-born players, even when they do not differ in terms of anthropometric and physical test data [17,18].

Thus, the peculiar ‘triple’ barrier for late-born soccer players consists of the primary selection stage, the growth spurt period, and the coaches’ subjective opinion. This barrier is the reason for the extremely low number of late-born soccer players and the over-representation of early-born soccer players in highly competitive youth teams aged between 16 and 18 years old [19]. However, young late-born soccer players are no less talented than their chronologically older peers, as this has been confirmed by many studies. These studies showed that late-born soccer players are more likely to reach the elite professional level and, at the very least, perform at this level just as successfully as early-born players [20,21,22].

The most commonly used explanation for this phenomenon is the ‘underdog’ theory that has been previously described [23,24]. According to this theory, late-born young athletes may gain an advantage through stronger psychological traits acquired in constant competition with physically stronger competitors. There is no doubt that this theory has a number of advantages, although it is also possible that the sporting success of late-born athletes at the elite adult level may have other explanations, which may be based on both socio-cultural and purely physiological factors.

In recent years, several studies evaluating the relationship between the relative age effect, biological maturity, anthropometry, and various parameters of physical performance have been published. Most of these studies have demonstrated that early-born young soccer players have an advantage in anthropometry and various parameters of physical performance compared to late-born players [25,26,27,28]. Altimari et al. showed statistically significant differences in stature and total and lean body mass but found no differences in lower limb power tests or aerobic fitness across young Brazilian soccer players of different birth tertiles. Although the authors found statistically significant results in RSA (repeated sprint ability) tests between the first and third birth tertiles in the U-13 and U-15 age groups [25]. Another study by Radnor et al. undertaken on soccer players from a male English academy presented no significant differences between each birth quartile in terms of weight, height, predicted adult height, and performance parameters, including 5 m, 10 m, 20 m, and 30 m sprint speeds and CMJ height. However, early-maturers were significantly taller and heavier compared with both on-time and late-maturers. From a performance aspect, early and on-time maturers significantly outperformed late-maturers in 5 m, 10 m, 20 m, and 30 m sprint times, but there were no differences in CMJ height between groups [29].

At the same time, there is a lack of data in the literature on this topic due to small statistical samples. Furthermore, these articles did not take into account soccer-specific skills such as ball dribbling, juggling, and passing accuracy.

In this regard, the aim of the study was to compare the status of somatic maturity, chronological age, anthropometry, strength, speed, and soccer-specific technical skills of young soccer players from leading youth soccer academies born in different quarters of the same calendar year. 

## 2. Materials and Methods

### 2.1. Study Type: Cross-Sectional Study (Level of Evidence 3)

Players from eight Russian Premier League academy youth teams were observed during the season. Of these players, 20% also represented the Russian national youth teams. The following anthropometric measures and physical characteristics were measured: height, weight, body mass index, countermovement jumps (CMJ), 5, 10, and 20 m sprints, speed dribbling, foot and body ball juggling, and short and long pass accuracy.

### 2.2. Subjects

Six hundred and seventy-eight male professional outfield soccer players (mean ± SD, 14.0 ± 1.4 years, height 174 cm ± 10.3 cm, weight 64.0 kg ± 11.0 kg, and BMI 20.9 ± 2.4) from eight Russian Premier League academy teams formed the sample. All data evolved as a result of employment, in which players were routinely monitored over the course of the competitive season. Nevertheless, approval for the study from the club was obtained, and ethics was approved by the Ethics Committee of Sechenov University (number 06–21 from 7 April 2021). 

The inclusion criteria for academies were as follows: selection by the national federation; presence of teams of all ages (from 6 to 18 years old); availability of a boarding school that hosts up to 40 players from other regions; and representation in youth national teams of at least five players.

The criteria for exclusion were as follows: an injury that caused the player to miss at least one of the three previous training days prior to the test; a period of less than 48 h from the end of an intense training session or game; no informed player or parent consent.

All dates of birth were divided into four quartiles according to the month of birth: January to March first quartile (Q1), April to June second quartile (Q2), July to September third quartile (Q3), and October to December fourth quartile (Q4). This classification has been previously employed in many studies involving soccer players [5,19].

According to the selection year in soccer, which runs from 1 January until 31 December, players born in the first quartile are considered early-born, and those born in the fourth quartile are considered late-born.

All data were analysed both in the total sample and after sub-dividing all players into birth quartiles and age group categories of 12–13, 14–15, and 16–17 years old.

### 2.3. Procedures

All subjects provided written informed consent and parental or guardian consent if required. To ensure confidentiality, all data were anonymised before analysis. 

All data were collected on the same day by the same testers to standardise procedures. All tests were carried out in the morning from 09:00 to 14:00, using the same equipment. Initially, all subjects were measured for standing height and body weight. A standardised breakfast two hours prior and a 15 min warm-up routine consisting of dynamic range of movement exercises were undertaken prior to each testing session. To exclude the influence of acute fatigue, a 3 min recovery after the warm-up routine was provided. Following the warm-up routine, the subjects then performed a 20 m sprint, all countermovement jumps, dribbling, ball juggling, and accuracy of short and long passing tests. A familiar standard-sized artificial turf soccer pitch was used for the testing protocol. All players wore their usual football equipment during testing. The use of new shoes was prohibited. All players were familiar with the battery of tests prior to the test start date and were provided the opportunity to perform trial runs. All subjects were familiar with the testing protocol as they were part of regular club assessments.

The percentage of predicted adult height determined by the Khamis-Rochet formula was used to determine maturity status. This method was proposed by Khamis et al. in 1994 and was first tested on white adolescents without pathological conditions residing in the USA [30]. This method of determining biological maturity is often used in various sporting organisations and in scientific research involving soccer players and can be considered the method of choice for determining maturity status [31]. 

### 2.4. Measurement of Standing Height and Body Weight

To measure the height of the players, a portable stadiometer model 217 manufactured by Seca (Gamburg, Germany) was used and placed on a solid, flat surface. All methods were performed in accordance with the relevant guidelines and regulations. Body-weight measurements were performed at the same time of day (before 10:00) in underwear and a T-shirt only and without shoes and socks.

The biological parents height was measured separately and under the strict supervision of the testing group. 

### 2.5. Sprint Test Protocol

Following the standardized warm-up routine, subjects performed two 20 m sprint trials on artificial grass. Between each of three attempts a 3 min recovery period was provided. The best single effort from the straight sprints and agility was used for analysis. 

Sprint times were recorded using a SmartSpeed Pro timing system (VALD Performance, Brisbane, Australia), with gates at zero, 5, 10, and 20 m [32]. This system uses a single-beam design to improve battery lifetime and ease of setup; it also incorporates novel error detection algorithms to reduce false triggers. In the event of multiple triggers, the algorithm interprets the longest trigger as the true start time. Gates were set at a height of 1 m from the floor. Each attempt was recorded with an accuracy of one hundredth of a second. Subjects started all sprint and agility trials from a two-point start position, with their front foot 0.3 m behind the first timing gate, and were instructed to complete with maximum effort. All tests were carried out in specific soccer shoes familiar to the players. 

### 2.6. Countermovement Jump (CMJ) Test Protocol

Following sprints, data on CMJ without armswing were collected. All participants were familiar with the jumping protocols due to their broad use as part of regular club assessments [33].

For each jump test, three attempts were made, and the best result was recorded in centimetres (cm) for further analysis. A recovery interval of 2 min between jumps was provided. A commercially available jump mat (SmartJump™, VALD Performance, Australia) was used and performed using the SmartJump system, which was previously validated. The subjects performed the tests in their normal soccer shoes on artificial grass.

### 2.7. Soccer-Specific Tests

The soccer-specific tests included dribbling, ball juggling, and short and long passes. These tests are part of the F-MARC test battery. They are conducted nationwide by the German football talent identification and development programme and are often used in testing soccer players due to their close relationship to standard soccer activity [34,35].

### 2.8. Speed Dribbling

The SmartSpeed Pro timing system (VALD Performance, Australia), six poles, and a ball were used to test dribbling. The testing protocol has previously been validated [35]. 

The player takes a standing position with his leg forward in front of the starting line; the body is tilted forward. The ball is located on the starting line of the timing gate. After stabilising the body position, the athlete begins to dribble the ball forward to the first pole with the maximum possible effort and circles the first pole on the left, the second pole on the right, and the third pole on the left. Next, he leads the ball to the fourth pole, circles it to the right, then the fifth pole to the left, and the last pole to the right (without knocking down the pole, although touching the posts is allowed). The player completes the test with maximum acceleration, with the ball crossing the finish line.

### 2.9. Juggling

The juggling test protocol was adopted from the F-MARC test battery [35]. The components of this test are described below.

#### 2.9.1. Juggling (Foot)

The player juggles the ball with his foot only, trying to touch the ball as many times as possible without letting it hit the ground. If 25 touches were achieved in the first attempt, no further attempts were made. To start, the ball is dropped to the foot by hand. The examiner measures the best of three attempts on the right and left sides. The measurement unit is 1 point per ball contact.

#### 2.9.2. Juggling (Body)

The examiner throws the ball from a 5 m distance to the player, who tries to touch and control the ball in the following order: (1) chest-foot-head, (2) head-left foot-right foot, (3) foot-chest-head. The examiner measures a total of three attempts per sequence. The measurement unit is 1 point per successful attempt. 

#### 2.9.3. Short Passing

The player dribbles the ball within a marked rectangle up to a line and, from there, passes accurately into a small goal 11 m away. The examiner measures a total of five attempts, scoring 3 points if a goal is scored and 1 point if the ball hits the crossbar or post [35].

#### 2.9.4. Long Passing

The player passes the ball from a static position into a circle (radius, 2 m; distance, 36 m) marked in the middle of a square target area (10 × 10 m). The player has one trial attempt. The examiner measures a total of five attempts. The measurement unit is 3 points if the ball lands in the circle or touches its circumference, and 1 point if the ball lands elsewhere inside the square [35].

### 2.10. Statistical Analyses

The data were collected in MS Excel and analysed in Jamovi 2.2.5. Descriptive statistical methods were used. The mean, SD, 95% confidence interval, minimum, and maximum were calculated. Normality was assessed using the Kolmogorov–Smirnov criterion. The chi-square test was used to assess the severity of the relative age effect (RAE), and the odds ratio and 95% confidence interval were calculated for 2 × 2 tables. ANOVA followed by Tukey’s post hoc test was used to assess differences in test parameters according to birth quartile, age group, and maturation percentage if statistically significant differences were found. Results were considered statistically significant at *p* < 0.05.

## 3. Results

There were 118 athletes in the 12–13-year-old age group, 322 in the 14–15-year-old age group, and 238 in the older 16–17-year-old age group. The normality of the population was confirmed using the Kolmogorov–Smirnov test.

The data revealed a high prevalence of the RAE in the examined sample. The number of early-born players is significantly higher than late-born players in both the overall sample and in each of the analysed age groups (Figure 1).

There was a high prevalence of the RAE in the overall group, with 43.5% of early-born players and only 9.6% of late-born players. There was no difference in the severity of the RAE between the 12–13- and 14–15-year-old age groups (*p* = 0.10, Chi = 6.24) and between the 14–15- and 16–17-year-old age groups (*p* = 0.13, Chi = 5.6). A significant difference was found between the 12–13- and 16–17-year-old age groups (*p* < 0.001, Chi = 16.7). The 12–13-year-old age group had significantly more late-born athletes than the 16–17-year-old age group (*p* = 0.002, OR 3.07 95% CI 1.48–6.37) and significantly fewer early-born players (*p* = 0.018, OR 0.36 95% CI 0.36–0.91).

The results of all conducted tests were analysed both in the total sample (Table 1) and after sub-dividing the group into birth quartiles and specific age groups (Table 2 and Table 3).

Post hoc analysis revealed a significant difference in age at months between Q1 and Q3 (*p* < 0.001), Q1 and Q4 (*p* < 0.001), Q2 and Q3 (*p* = 0.008), and Q2 and Q4 (*p* = 0.002) soccer players. 

First-quartile players weighed more than Q3 and Q4 players (*p* = 0.003 and *p* = 0.018, respectively), and Q2 players weighed more than Q3 players (*p* = 0.041). 

The height of early-born players (Q1) was significantly higher than that of Q3 and Q4 players (*p* < 0.001 and *p* = 0.008, respectively), while Q2 players were higher than Q3 and late-born players (Q4) (*p* = 0.004 and *p* = 0.045, respectively).

When analysing each of the age groups individually (12–13, 14–15, and 16–17 years), there was no statistically significant difference in anthropometrical measures or physical or technical tests between birth quartiles. 

When comparing the degree of maturation of players from different quartiles, findings indicated that early-born players (Q1) were more mature than Q3 and ‘late-born’ players (Q4) (*p* < 0.001 and *p* < 0.001), while Q2 players were more mature than Q3 and Q4 players (*p* = 0.016 and *p* = 0.007).

Analysis of the data obtained using ANOVA showed that athletes born in different quartiles differed in maturity, age, height, and weight, but no statistically significant differences in strength, speed, or sport-specific skills were observed.

At the same time, players from different age groups differed significantly in both maturity status and anthropometry, strength, speed, and sport-specific skills, where it was suggested that players from the older age group had an advantage.

## 4. Discussion

The results of the study showed that early-born soccer players have higher maturity status and anthropometric measures than their late-born peers. Nonetheless, these differences do not determine the better development of speed, strength, and sport-specific skills, such as dribbling. Therefore, the presence of late-born players in the teams of leading academies can be associated not only with their higher maturity status (comparable to early-born), but also with greater physical and technical talent. However, the number of such players is much lower due to widespread RAE among 12- to 18-year-old players in elite soccer academies. The effect is especially prevalent in the 16–17 age group. The number of early-born players was significantly higher in this group, while the number of late-born players was significantly lower when compared to the younger group (12–13 age group). These findings are in line with previous studies examining young players from other leading European academies [36,37]. These data demonstrate that the selection process during the growth spurt was performed according to the maturation gradient and may be based on a dramatic increase in the serum concentration of testosterone, the level of which does not differ between the sexes or ages of 6 to 10 years, but rises rapidly in boys around 12 years of age, which may account for the sudden improvement in sports performance at this age [14]. 

According to Senefeld et al., the mean testosterone concentration in boys aged 6 to 20 years increases from 0.07 nmol/L to 17.9 nmol/L at age 20 years (*p* < 0.001), with a plateau beginning at age 17 years [38]. In these circumstances, young football players with an early growth spurt can gain an advantage in sports. A recent study confirmed the influence of biological maturity on the success of 14–15-year-old young soccer players, in which a significant relationship was demonstrated between the level of biological maturity and key parameters of physical match performance such as peak speed, distance covered at high speed, performance in a 40 m sprint, and the CMJ [39]. It is important to note that the positive effect of increased testosterone levels on athletic success may be related not only to the effects on physical parameters, but also to a range of psycho-emotional factors such as motivation, focus on success, and dominance [40].

With regard to the influence of birth quartile on the physical condition and technical skills of young soccer players, the evidence is currently conflicting. Several studies have provided evidence to support a positive relationship between relative age and physical test scores [25,41,42]. Other studies have shown that there was no statistically significant difference between players from different birth quartiles, either in anthropometric parameters or in various technical and physical skills [42,43]. A large study by Votteler et al. involving more than 10,000 soccer players aged 12–15 showed that revised motor performance diagnostics highlighted improved average results for younger players, and path analysis revealed significant RAE during physically demanding tests, although almost no effect for technically demanding tests [37]. However, only a small number of these studies assessed biological maturity status, which therefore does not fully explain the results of this research. In those studies where maturity status was assessed, the degree of biological maturity rather than birth quartile was the key factor influencing test results, and the authors concluded that a higher biological maturity status for late-born soccer players allowed this study group to compete with early-born players. However, physical and technical parameters are not always strictly related to maturity status. For example, in a study by Figueiredo et al. examining a large sample of elite young soccer players aged 11–14 years old, it was shown that early-born players were heavier and taller than late-born players; however, maturity status variation did not alter functional capacity, soccer-specific skills, or goal orientation [44].

However, the most important finding of the present study is that late-born (Q4) young soccer players are significantly statistically different in chronological age and less mature than early-born (Q1) players. These Q4 players also have lower height and weight than early-born players, although no differences in physical and technical performance were evident, and thus have comparable technical skills to the early-born players. 

The present findings may suggest that the sporting success of late-born soccer players at the adult level may be based not only on the underdog effect, but also on the greater talent of late-born young soccer players who managed to remain on the most competitive teams. In such a situation, the lower maturity status against a background of comparable physical and technical skills (dribbling, juggling, short and long passes) of late-born players may provide them with an advantage in the future. It can be assumed that the sporting success of late-born soccer players in adult professional elite soccer is based on a number of complex factors, including both an underdog effect and greater talent in such players. This probably contributed to their ability to stay on highly competitive youth soccer teams during one of the most challenging periods of growth and maturation at the age of 12–16. Furthermore, the intention was to gain an advantage at 17–18 years old, which is regarded as the most important age category to ensure the successful transition into adult professional soccer.

However, players born in the same year should not be considered to have more or less talent solely based on their birth quartile. It can be assumed that under the current selection system, less mature and less anthropometrically developed players may have an opportunity to stay in elite youth soccer only if they have comparable physical condition and technical skills with their more mature peers. These players are scarce in youth soccer; nevertheless, they have the potential to gain an advantage in adult soccer. Although the present study showed that early-born soccer players have a higher maturity status than late-born players, it should also be noted that the RAE and biological maturity need to be considered independent constructs. Apparently, among children born in each of the twelve months, there are both early-, on-time-, and late-maturing children, and the degree of their physical and cognitive development is comparable with their maturity status [45,46].

This study is one of the first to compare not only anthropometry and traditional tests of speed and strength (sprints and jumps), but football-specific skills such as ball dribbling, juggling, and passing accuracy on a large sample of various age players from elite soccer academies. This makes the obtained data interesting both for scientists, coaches, and scouts.

The first limitation of this study is the lack of a cognitive skills assessment, whose development may not match the development of physical qualities and technical skills. This is considered one of the limiting factors in dividing soccer players into groups according to maturity rather than chronological age, commonly called bio-banding. It is believed that children of similar maturity but different chronological ages may have different degrees of cognitive development, which reduces the potential usefulness of bio-banding as a way to reduce selection bias in soccer. 

The second limitation is the use of a percentage of the predicted adult height as a means of assessing maturity status. This method has been frequently employed in youth soccer and has previously been shown to have an acceptable correlation with various methods of determining maturity status, such as X-rays and ultrasound. However, the validity of the percentage of predicted adult height method must be considered further, as it depends on an accurate measure of the athlete’s height and the height of the biological parents. Additionally, the hand radiography method to determine skeletal age is still regarded as the gold standard method to determine biological maturity. 

Further research should focus on examining the range of factors responsible for the retention of late-born players in elite soccer and their role in different periods of their sporting careers. The question of the underdog hypothesis in the most competitive soccer organisations still remains unexplored. For example, do late-born players in competition with taller and heavier early-born players really become mentally stronger and psychologically more stable? Or are the few late-born players who have managed to stay in elite youth soccer initially more talented? It is also highly relevant to investigate measures to reduce the prevalence of the RAE in the most competitive soccer academies, both at the primary selection stage and at the selection stage between the ages of 12 and 15 years.

## 5. Conclusions

The number of players born in Q4 was significantly under-represented; they are less mature than players born in Q1, but do not differ in their physical and technical performance in the examined leading soccer academies.

These findings suggest that the sporting success of late-born players in elite adult soccer may be based not only on the underdog hypothesis, but also on their greater talent.

## Figures and Tables

**Figure 1 sports-11-00179-f001:**
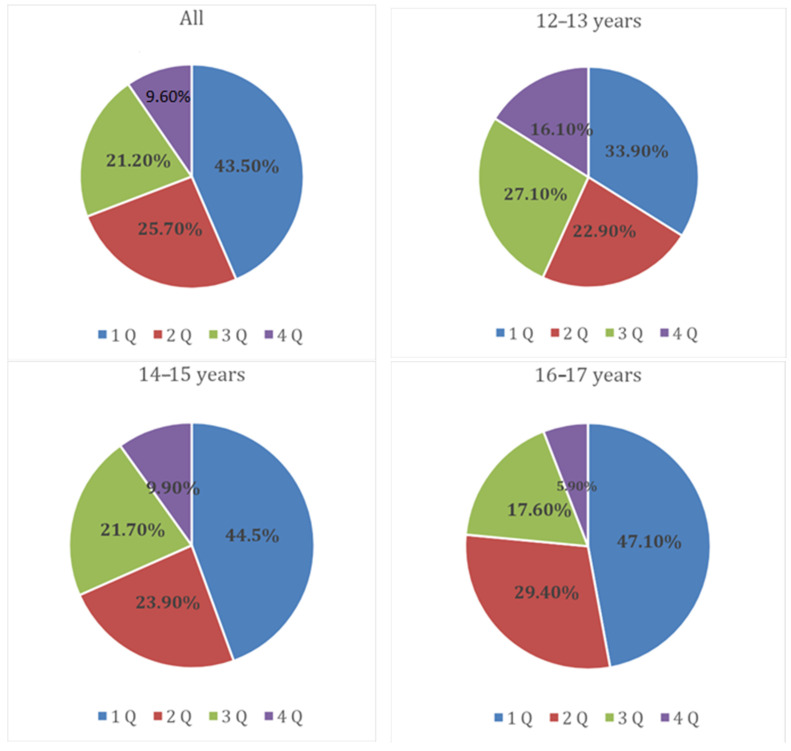
Prevalence of the RAE in young soccer players in the overall sample and in different age groups.

**Table 1 sports-11-00179-t001:** Physical and technical data from players aged 12–17 years old.

Parameters Analysed	Minimum	Maximum	Average	Standard Deviation
Maturation, %	72.60	100	94.60	4.70
CMJ, cm	24.00	54.10	38.40	5
Age, years	12.10	17.90	14.00	1.40
Height, cm	116	199	174	10.30
Weight, kg	32.60	96.40	64	11
Age, months	146	215	184	16.60
Sprint 5, m/s	0.88	1.28	1.09	0.06
Sprint 10, m/s	1.60	2.12	1.83	0.09
Sprint 20, m/s	2.78	3.71	3.13	0.16
Dribbling, s	8.10	12.20	9.70	0.60
Juggling, points	0	50	45.90	8.05
Short passes, points	0	15	9.70	3.21
Long passes, points	0	15	6.20	2.90

**Table 2 sports-11-00179-t002:** Physical and technical data from players aged 12–17 years old born in different quarters. CI is included in brackets in the table cells.

	Q1 (n = 295)	Q2 (n = 174)	Q3 (n = 144)	Q4 (n = 65)	ANOVA *p*-Value
Maturation, %	95.50 (95.10–96)	94.90 (94.20–95.50)	93.30 (92.40–94.20)	92.70 (91.40–94)	<0.001
CMJ, cm	38.80 (38.20–39.30)	38.60 (37.80–39.30)	37.90 (37.10–38.70)	37.80 (36.4–39.20)	0.25
Age, months	187 (185–188)	186 (183–188)	180 (177–182)	177 (173–181)	<0.01
Age, years	15	15	14.60	14.30	<0.001
Weight	65.40 (64.30–66.60)	64.80 (63.10–66.50)	61.50 (59.60–63.40)	61 (58.30–63.80)	<0.001
Height	1.76 (1.75–1.77)	1.75 (1.74–1.77	1.72 (1.70–1.73)	1.72 (1.69–1.74)	<0.001
Sprint 5, m/s	1.08 (1.07–1.09)	1.08 (1.07–1.09)	1.09 (1.08–1.10)	1.09 (1.08–1.11)	0.44
Sprint 10, m/s	1.82 (1.81–1.83)	1.83 (1.81–1.84)	1.83 (1.82–1.85)	1.84 (1.82–1.87)	0.34
Sprint 20, m/s	3.12 (3.10–3.14)	3.13 (3.11–3.16)	3.15 (3.12–3.18)	3.18 (3.13–3.22)	0.12
Dribbling, s	9.72 (9.65–9.79)	9.71 (9.61–9.80)	9.63 (9.52–9.74)	9.67 (9.53–9.81)	0.30
Juggling, points	44.80 (43.80–45.90)	46.9 (46.00–47.90)	46.80 (45.60–47.90)	45.80 (43.90–47.60)	0.06
Short pass, points	9.92 (9.56–10.30)	9.64 (9.16–10.10)	9.53 (9.04–10)	9.14 (8.25–10)	0.29
Long pass, points	6.21 (5.89–6.54)	6.22 (5.80–6.64)	6.24 (5.74–6.73)	6.18 (5.49–6.88)	0.98

**Table 3 sports-11-00179-t003:** Physical and technical data from players of different age groups. CI is included in brackets in the table cells.

	12–13 Years (n = 118)	14–15 Years (n = 322)	16–17 Years (n = 238)	ANOVA *p*-Value
Maturation, %	86.90 (86.20–87.50)	94.40 (94.10–94.70)	98.60 (98.40–98.70)	<0.001
Weight	50.20 (48.60–51.70)	63.80 (62.90–64.80)	71.10 (70.10–72.10)	<0.001
Height	1.62 (1.60–1.64)	1.75 (1.74–1.76)	1.79 (1.78–1.80)	<0.001
CMJ, cm	34.40 (33.70–35.10)	38.50 (38–39.10)	40.30 (39.80–40.90)	<0.001
Age, years	12.70 (12.60–12.70)	14.50 (14.50–14.60)	16.40 (16.30–16.50)	<0.001
Age, months	159 (158–160)	180 (179–181)	202 (201–203)	<0.001
Sprint 5, m/s	1.14 (1.13–1.16)	1.08 (1.07–1.09)	1.06 (1.06–1.07)	<0.001
Sprint 10, m/s	1.93 (1.91–1.95)	1.82 (1.81–1.83)	1.78 (1.77–1.79)	<0.001
Sprint 20, m/s	3.35 (3.32–3.38)	3.13 (3.11–3.03)	3.04 (3.03–3.05)	<0.001
Speed dribbling, s	10.0 (9.91–10.1)	9.68 (9.61–9.74)	9.55 (9.47–9.63)	<0.001
Ball juggling, points	44.5 (43.1–46.0)	45.5 (44.6–46.5)	47.0 (46.2–47.8)	0.007
Short pass, points	9.01 (8.49–9.53)	9.5 (9.14–9.86)	10.3 (9.89–10.7)	<0.001
Long pass, points	4.22 (3.70–4.74)	6.58 (6.29–6.87)	6.71 (6.37–7.06)	<0.001

## Data Availability

The data presented in this study is not publicly available and can be obtained via request from the corresponding author.

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
