# Peer review of "Are Late-Born Young Soccer Players Less Mature Than Their Early-Born Peers, Although No Differences in Physical and Technical Performance Are Evident?"

_sports, 2023, doi:10.3390/sports11090179_

Round 1

Author Response

We are grateful for your valuable commentaries and great suggestions. We did our best to fix them as soon as possible. Point-by-point responses are included within each comment following a dashed line.

Reviewer 1

This study aimed to clarify the relative age effects on anthropometry and physical

performance in youth soccer players. The issue is important and significance for sports

scientists and players. However, there are some revisions for publication.

[Major comments]

  1. What’s the novelty and originality of this study? In preadolescence and adolescence,

relative age effects (RAE) exist in different sports. Please describe the rational

explanation in the introduction section. Little information is found on the characteristics

of anthropometry and physical performance relevant to maturity and RAE. – We included a paragraph about novelty of study in introduction section.

  1. Why did authors provide the information on the birth- and chronological age-related

differences in the measured variables? It is necessary to indicate what the author want

to say from the two results in the text.  – Dear reviewer! We tried our best to rewrite our discussion section according to your precious remarks. Unfortunately, we didn't fully understand this commentary. If you are still dissatisfied with our article, please explain in more detail and indicate the lines. We are very grateful for your help and time.

  1. In my opinion, the RAE would be smaller in the older group than in the younger group.

Therefore, it is better to analyze birth-related difference in the measured variables within

the same age group. – Our data show the opposite. A paragraph from results section: “A significant difference was found between the 12-13- and 16–17-year-old age groups (p<0.001, Chi = 16.7). The 12–13-year-old age group had significantly more late-born athletes than the 16–17-year-old age group (p=0.002, OR 3.07 95% CI 1.48-6.37) and significantly fewer early-born players (p=0.018, OR 0.36 95% CI 0.36-0.91).”

[Minor comments]

  1. Discussion. Body size is partly related to physical performance. In this study, the birth-related difference in body size were found. Why? – We reworked our discussion section according to your remark.
  2. Table 2. The values of “Age, year” and “Age, month” are reversed. - fixed

Reviewer 2 Report

The idea of this study is interesting, my recommendations are the following:

In the abstract - measuring height and body weight are not physical tests, they are anthropometric measurements, I recommend correction.

Keywords - I recommend replacing the keyword underdog hypothesis, it does not appear in the study.

Since in the introduction you refer to the football selection, my recommendation is to change the title and insert the word selection.

Lines 90-91 in the abstract mention other physical tests, I recommend clarification.

Line 98 I recommend that you mention the registration number of the ethics agreement.

I recommend moving lines 101-113 to the Procedure section.

I recommend that the inclusion and exclusion criteria be mentioned as phrases. The arrangement of this article is not the most efficient.

Line 125 I recommend rewriting, it is understood that the other tests were not in the same conditions.

Lines 126-131 recommend moving to the Procedure section.

I recommend reorganizing the Subjects section, it is very difficult to understand, go from one aspect to another.

Lines 140-141 repeat the idea, I recommend deleting it.

Table 2 - the last column you mention ANOVA, which parameter are you referring to???

I recommend that in the tables you use the classical formulation, for example, the arithmetic mean is entered with the ± sign, also two decimal places are used, please change it. The arithmetic mean is entered in the same column as the standard deviation.

Table 1 - average age, arithmetic average of height and weight were also mentioned in the Subjects subsection, I recommend deleting them.

I recommend joining table 1 with 3, and rearranging it.

Table 3 - what the brackets represent, I don't consider the minimum and maximum to be relevant.

I recommend rewriting the interpretation of the data, without duplicating the information.

Lines 281-286 are not relevant, this is informative, it was not targeted in the study.

Lines 328-336 recommend deletion, double the information mentioned in the Subjects or Procedure section.

In the Discussions section, correlations are made between the data of the present study and data from previous studies. In this idea, I recommend that the entire section be rethought and rewritten.

As a final conclusion this article should be reorganized because certain sections are confusing and other sections rewritten. The statistics are very poor.

Author Response

We are grateful for your valuable commentaries and great suggestions. We did our best to fix them as soon as possible. Point-by-point responses are included within each comment following a dashed line.

Reviewer 2

The idea of this study is interesting, my recommendations are the following:

In the abstract - measuring height and body weight are not physical tests, they are anthropometric measurements, I recommend correction. – Corrected.

Keywords - I recommend replacing the keyword underdog hypothesis, it does not appear in the study. - Removed

Since in the introduction you refer to the football selection, my recommendation is to change the title and insert the word selection. – We would like not to change the title.

Lines 90-91 in the abstract mention other physical tests, I recommend clarification. – List of all physical tests is included.

Line 98 I recommend that you mention the registration number of the ethics agreement. – Number is added.

I recommend moving lines 101-113 to the Procedure section. – Lines are moved

I recommend that the inclusion and exclusion criteria be mentioned as phrases. The arrangement of this article is not the most efficient. – Fixed.

Line 125 I recommend rewriting, it is understood that the other tests were not in the same conditions. – Sentence is corrected.

Lines 126-131 recommend moving to the Procedure section. – Lines are moved.

I recommend reorganizing the Subjects section, it is very difficult to understand, go from one aspect to another. – This section is reorganized according to your recommendations, thank you.

Lines 140-141 repeat the idea, I recommend deleting it. – Corrected. “First born” and “late-born” are removed in the brackets.

Table 2 - the last column you mention ANOVA, which parameter are you referring to??? – We are referring to p-value. We included “ANOVA p-value” in header of every table.

I recommend that in the tables you use the classical formulation, for example, the arithmetic mean is entered with the ± sign, also two decimal places are used, please change it. The arithmetic mean is entered in the same column as the standard deviation.

Table 1 - average age, arithmetic average of height and weight were also mentioned in the Subjects subsection, I recommend deleting them. – We would like to present united information in this table.

I recommend joining table 1 with 3, and rearranging it. – Regrettably, we can’t join them, because they represent different information (maximum, minimum and standard deviation in table 1, and CI in table 3)

Table 3 - what the brackets represent, I don't consider the minimum and maximum to be relevant. – Brackets represent CI, we consider that relevant information.

I recommend rewriting the interpretation of the data, without duplicating the information. – We tried our best to remove duplicates.

Lines 281-286 are not relevant, this is informative, it was not targeted in the study. – In our opinion, this information is important as it elucidates the potential advantage of older children.

Lines 328-336 recommend deletion, double the information mentioned in the Subjects or Procedure section. – Great point, this paragraph is removed.

In the Discussions section, correlations are made between the data of the present study and data from previous studies. In this idea, I recommend that the entire section be rethought and rewritten. – We rewrote and reworked the discussion section.

As a final conclusion this article should be reorganized because certain sections are confusing and other sections rewritten. The statistics are very poor. – We tried  our best to reorganize the article according to your valuable review.

Reviewer 3 Report

Dear authors, 

Having reviewed the research, I consider it to be evidence of high novelty. However, the following changes are suggested: 

The research needs more contextualisation in the theoretical section of the introduction. 

The whole article should be written in impersonal, avoiding the use of "we" and "our".

Provide a table with the demographic results of the sample. 

The normality of the population has not been studied. I recommend using the Kolmogorov-Smirnov test. 

There are some tests for data collection that have not been used by the scientific community. They need to be justified theoretically. 

The discussion needs to be expanded

English should be improved. I recommend further reading

Author Response

We are grateful for your valuable commentaries and great suggestions. We did our best to fix them as soon as possible. Point-by-point responses are included within each comment following a dashed line.

Reviewer 3

Dear authors, 

Having reviewed the research, I consider it to be evidence of high novelty. However, the following changes are suggested: 

The research needs more contextualisation in the theoretical section of the introduction. – We included a paragraph about novelty of study.

The whole article should be written in impersonal, avoiding the use of "we" and "our". – The article is now impersonal.

Provide a table with the demographic results of the sample. – The number of athletes in each age group and their mean age are already included in results section.

The normality of the population has not been studied. I recommend using the Kolmogorov-Smirnov test. – We used this test to confirm normality of the population (it was described in statistical analyses). We also included “The normality of the population was confirmed using Kolmogorov-Smirnov test.” in results section.

There are some tests for data collection that have not been used by the scientific community. They need to be justified theoretically. – All tests are widely known. Each test in methods section includes references.

The discussion needs to be expanded – We rewrote and reworked the discussion section.

Round 2

Author Response

Dear Reviewer! Please accept our deepest gratitude for your valuable thoughts and comments.

Author stated to describe the novelty of this study in the introduction section. The earlier studies reported about relative age effect have been already published by other groups. What do we know from previous findings? Further, please add a detailed description of what the problem is and how it was solved based on what we know so far in the revised text.  - We included few paragraphs that includes data from previous studies in introduction section. 
2. Author have tried to clarify the impact of relative age effect on anthropometry and physical performance. Why did you also present the result of chronological age-related differences in anthropometry and physical performance, when you should have presented only the results of birth-related differences? - We presented these results for two reasons. 1. One of the reviewers from previously submitted Q1 sports medicine journal strongly recommend to include this table. 2. This table demonstrates the presence of differences in anthropometrical measures, physical and technical tests across the selected age groups. We justified including this information because the reader may question if the absence of the mentioned differences across birth quartiles, in the first place, is caused by their absence across age groups.
3. Sorry that you did not understand the intent of my question. I have understood the results of the relative distribution of birth month. You have presented the results of birth-related differences in anthropometry and physical performance across chronological age. Because the chronological age may influence on the birth-related differences in anthropometry and physical performance, you should better examine these differences within the same age group. - We did examine these differences, which is now clearly mentioned in the reworked paragraph in the results section: "When analysing each of the age groups individually (12-13, 14-15, 16-17 years), there was no statistically significant difference in anthropometrical measures, physical and technical tests between birth quartiles". We decided not to include another table because of it's excessive size and the absence of statistically significant results.

We hope that you will be satisfied with this revised version of article.

Best regards,
Authors of the article.

Reviewer 2 Report

No comments

Author Response

Thank you!

Reviewer 3 Report

The paper has been improved.

Author Response

Thank you!